# “Ferrocrinology”—Iron Is an Important Factor Involved in Gluco- and Lipocrinology

**DOI:** 10.3390/nu14214693

**Published:** 2022-11-06

**Authors:** Michał Szklarz, Katarzyna Gontarz-Nowak, Wojciech Matuszewski, Elżbieta Bandurska-Stankiewicz

**Affiliations:** Clinic of Endocrinology, Diabetology and Internal Medicine, School of Medicine, Collegium Medicum, University of Warmia and Mazury in Olsztyn, 10-957 Olsztyn, Poland

**Keywords:** Ferrocrinology, myocrinology, metabolism, obesity, diabetes

## Abstract

“Ferrocrinology” is the term used to describe the study of iron effects on the functioning of adipose tissue, which together with muscle tissue makes the largest endocrine organ in the human body. By impairing exercise capacity, reducing AMP-activated kinase activity, and enhancing insulin resistance, iron deficiency can lead to the development of obesity and type 2 diabetes mellitus. Due to impaired browning of white adipose tissue and reduced mitochondrial iron content in adipocytes, iron deficiency (ID) can cause dysfunction of brown adipose tissue. By reducing ketogenesis, aconitase activity, and total mitochondrial capacity, ID impairs muscle performance. Another important aspect is the effect of ID on the impairment of thermogenesis due to reduced binding of thyroid hormones to their nuclear receptors, with subsequently impaired utilization of norepinephrine in tissues, and impaired synthesis and distribution of cortisol, which all make the body’s reactivity to stress in ID more pronounced. Iron deficiency can lead to the development of the most common endocrinopathy, autoimmune thyroid disease. In this paper, we have discussed the role of iron in the cross-talk between glucocrinology, lipocrinology and myocrinology, with thyroid hormones acting as an active bystander.

## 1. Introduction

Iron is essential for proper functioning of all cells in the human body. It is involved in the transfer of oxygen and electrons and the activity of numerous enzymes [1]. Iron deficiency is among the most prevalent nutritional deficiencies, affecting as many as two billion people all over the world, mostly pregnant women and children [2]. Its prevalence to much extent depends on socioeconomic status. It affects 4–18% of the population in the United States and Northern and Western Europe, 9–50% of the population in Eastern Europe, 64% in Asia, 54% in southern Asia and 62% in Latin America [3]. Adequate iron supply is essential for the proper metabolism of endocrine organs.

The term glucocrinology was coined by Karl et al. in 2018 to name the study of the relationship between diabetes and endocrine disorders. Glucocrinology includes in its ambit endocrinopathies that may cause secondary diabetes [4], including autoimmune thyroid disease (AITD), which is the most common endocrinopathy in the world. Lipocrinology studies the relationship between fatty tissue metabolism and endocrine system functions as well as the fatty tissue as an endocrine organ [5]. 

The aim of this paper is to discuss the role of iron as a new link between glucocrinology, lipocrinology and thyrometabolism. 

## 2. Introduction to Ferrocrinology—Effects of Iron Deficiency on Thermoregulation and Stress Response

The first studies of the effects of iron on thyroid function focused on thermoregulation mechanisms, which, when impaired, may underlie metabolic disorders in iron deficiency. Defects of these mechanisms in iron deficiency (ID) states have been shown in a number of studies in animals and humans. ID can lead to thermoregulation disorders in two mechanisms—through anaemia and through tissue iron deficiency. Anaemia impairs the production and conservation of heat, while tissue iron deficiency affects the sympathetic and endocrine systems, impairing body response to cold exposure and thermogenesis. Brown adipose tissue (BAT) is specialized fat tissue with high capacity to dissociate cellular respiration from adenosine 5’-triphosphate (ATP). BAT activity in humans is inversely correlated with adiposity, blood glucose concentrations, and insulin sensitivity [6].

### 2.1. ID Impairs Hormonal Response to Hypoxia

In iron deficiency states, hypoxia induced by a decrease in oxygen concentrations (down to 10 and 50% of baseline values) more rapidly led to hypothermia [7] and decreased oxygen transport into cells [8]. Hypoxia in iron-deficient rats resulted in impaired vasoconstriction at cold temperatures, decreased shivering and non-shivering thermogenesis, and a lower core body temperature set-point [9]. Sudden exposure to hypoxia (equivalent to 5 km above sea level) resulted in decreased thyroid hormone production. Moreover a decrease was observed for both iodine uptake and its total amount in the thyroid gland. Dysfunction of the thyroid gland in rats persisted for up to 30 days after the exposure to hypoxia. Body mass loss and reduced radioiodine I-131 uptake were observed as early as on day 3 and persisted for 30 days. Moreover there was a decrease in tetraiodothyronine (T4) levels and a decrease in metabolism and plasma clearance of T4 (due to reduced deiodinase activities). The T4 deiodination index and the rate of iodine extraction from the thyroid gland were markedly reduced due to the reduced activity of thyroglobulin proteolysis in hypoxia. In hypoxia there is also a decrease in thyreotropin (TSH) secretion [10], and chronically hypoxic children had lower T4 and triiodothyronine (T3) levels with concomitant increase in reverse triiodothyronine (rT3) levels [11].

Similar to environmental hypoxia is the hypoxia induced by anaemia. Therefore the results of studies in which anaemia impairs the ability to maintain normal body temperature upon exposure to the cold should not be surprising. In a porcine model of anaemia (haematocrit reduced from 26% down to 15%), Mayfield et al. have found that animals with anaemia remained longer in hypothermia, and their ability to increase oxygen consumption at cold exposure remained impaired for a longer time [12]. In the rat study, the group with reduced haematocrit did not achieve normal body temperature and did not increase their TSH, free triiodothyronine (fT3), and free tetraiodothyronine (fT4) secretion. Increase in haematocrit up to 80% of normal restored normal thyroid response to the cold at 6 days following the intervention [13].

### 2.2. Iron Deficiency Impairs the Utilization of Norepinephrine and Favours the Hyperadrenergic State 

Interscapular brown adipose tissue (IBAT) responsible for non-shivering thermogenesis is rich in mitochondria and is heavy innervated by sympathetic nerves. Upon cold exposure it is getting more blood supply. Norepinephrine binds to alpha and beta receptors in the IBAT and initiates the cascade of reactions resulting in an increase in mitochondrial oxidative phosphorylation and up-regulation of deiodinase 2, synthesis of which depends also on availability of T4. The adrenergic system, in interaction with thyroid hormones, is of key importance for thermogenesis. In cold temperatures, more norepinephrine is released and bound to adrenergic receptors. The sympathetic nervous system at cold exposure is stimulated from skin receptors, the brainstem, and the spinal cord. The adrenergic response is integrated in the preoptic/anterior hypothalamic (POAH) area, which contains thermosensitive neurons [14]. Thyrotropin-releasing hormone (TRH) administration was shown to increase norepinephrine levels and stimulate catecholamine metabolism [15]. The sympathetic stimulation can alter blood flow, resulting in a decrease in heat loss and an increase in heat production via its effects on the brown adipose tissue. T4 production in the thyroid gland is increased at exposure to cold. However T3 production is increased even more (via the conversion of T4 to T3 by 5′-deiodinase), resulting in the saturation of its nuclear receptor [16]. Noradrenergic receptor stimulation upregulates the 5′deiodinase mRNA expression, thereby increasing the conversion of T4 to T3 (while T3 is approximately tenfold more potent than T4). Thyroid hormones increase the sensitivity to norepinephrine (NE) via the interaction between beta-adrenergic receptor and adenyl cyclase [17]). T3 also exerts systemic effects resulting in the acceleration of metabolism (with increased Na/K ATPase activity and intracellular calcium levels) [14]. 

Another problem is impaired neuronal control of thermoregulation in the case of iron deficiency. Hypothyroidism leads to down regulation of beta receptors in brown adipose tissue and alpha-1 and alpha-2 receptors in the liver, which reflects the compensatory hyperadrenergic state [18]. 

In iron deficiency states, sympathetic tone is elevated, while norepinephrine levels are increased in some tissues and decreased in blood and urine [19]. As previously mentioned, this may result from a kind of compensation for reduced T3 levels in tissues [20]. Alterations in catecholamine breakdown in cases of iron deficiency were confirmed in rat studies [21], while in other studies rats had considerably increased norepinephrine levels in blood and urine [13,22]. This effect was more pronounced in low temperatures, and the effect of ID was reversible upon iron supplementation, but did not correlate with anaemia correction [22]. Smith et al., who treated animals with a ganglionic blocker chlorisondamine, have found a considerably more rapid increase in norepinephrine levels in hearts of iron deficient rats [23]. In a study by Beard et al., plasma clearance of norepinephrine in ID was considerably lower, with impaired turnover of norepinephrine in tissues. Iron supplementation resulted in a sixfold increase in norepinephrine metabolism [24]. In other studies, iron deficient rats were unable to maintain normal body temperature during their exposure to 4 °C and showed lower thyroid hormone levels with elevated norepinephrine levels [25,26]. ID can also affect muscle ability to use energy for muscle contractions, which impairs shivering thermogenesis [27]. Studies in humans also confirm the contributory effect of iron deficiency on increased norepinephrine levels [28,29]. A study by Voorhees et al. has shown elevated catecholamine levels in children with iron deficiency [28].

### 2.3. ID Impairs TSH Secretion

Exposure to cold is expected to increase TRH and TSH secretion. As already mentioned, the endocrine system stimulation in response to low temperatures is mediated by POAH neurons. In a study by Sato et al. from 1985, TRH administration resulted in hyperthermia in rats [15]. Beard, Green, Miller, and Finch observed no increase in TSH levels after 6 and 24 h of cold exposure in iron deficient rats [30]. In the group of rats with iron deficiency TSH, fT3 and fT4 levels were reduced and their response to cold at 1 and 5 h, which should be reflected in TSH increase, was impaired [31]. This blunted response to the cold in iron deficiency may depend on dysfunction of the hypothalamic-pituitary-thyroid axis. Tang et al. measured TSH levels in response to exogenous TRH and found that pituitary response to TRH was not altered but compensatory TRH secretion in the hypothalamus was disturbed, which failed to provide adequate TSH concentrations in cold temperatures [31]. Suppression of hypothalamic TRH secretion in iron deficiency states may result from increased dopamine release in neuronal pathways of the ventral midbrain. Iron deficient rats were shown to have increased levels of extracellular dopamine in the caudate putamen [32] and a reduced number of dopamine D2 receptors [33]. Rats with ID can develop dopamine-dependent sleep-wake disorders [34]. D2 receptor down-regulation can attenuate the dopamine-mediated thermogenic effect. Secondly, the reduced number of D2 receptors can result in lower hypothalamic secretion of TRH, with subsequent blunting of the normal response to cold, while increased presynaptic dopamine levels may cause hypothermic effects [14]. However some other studies have shown delayed pituitary response to TRH secretion at a haematocrit level of 16% versus control group [35]. The opposite previous findings by Tang et al. can be explained by higher haematocrit (30%) in their studied individuals [31]. Control at the pituitary level may also be impaired. The conversion of T4 to T3 in the pituitary gland depends on the outer ring of 5’deiodinase, which is a key rate-limiting factor in the control of TSH secretion, while in iron deficiency this extremely important regulation may be disturbed [19]. In haemodialysis patients with anaemia, the TSH response to TRH administration was delayed, which most probably was caused by too high sensitivity of pituitary thyrotropes to T3 and T4 inhibition or by transient decrease in blood supply to this gland. Upon correction of anaemia, normal TSH secretion in response to TRH was restored [36]. In another study with human subjects, an increase in TSH, fT4, and fT3 was reduced by 4–7% in women with iron deficiency [37]. 

### 2.4. ID Results in Impaired Binding of Triiodothyronine to Its Nuclear Receptor 

Higher T3 nuclear receptor saturation along with adrenergic stimulation upregulate the mRNA of uncoupling proteins and result in heat production. BAT is highly sensitive to T3 because the amount of T3 nuclear receptors in BAT is similar to that in the pituitary gland and liver. Moreover, the adrenergic stimulation can increase the activity of lipoprotein lipase, an enzyme responsible for the release of free fatty acids (FFA) from triglycerides (TAG), which activate proton translocation across the inner mitochondrial membrane [14]. The ability to accelerate oxidative processes in BAT depends on thermogenin, an uncoupling protein (UCP) found in the inner mitochondrial membrane. Rapid exposure to cold upregulates the expression of thermogenin. T3 is crucial for the responsiveness to UCP, while 5′deiodonase is crucial to maintain sufficient levels of T3 in the tissues. The activity of 5′deiodinase is directly proportional to T3 levels. It is UCP that makes excess energy dissipation possible. What is interesting is that the relationship between UCP response and T3 nuclear receptor occupation is not linear. The 70–75% receptor occupation in the temperature of 25 °C in euthyreosis suggests that an adequate IBAT response to cold requires even higher receptor saturation [38].

These preliminary findings were confirmed in animal studies by Silva et al., who showed that appropriate non-shivering thermogenesis requires 90% saturation of T3 nuclear receptor [39]. Iron deficiency impairs T3 binding to its nuclear receptor at cold exposure [40]. An indicator of adequate intracellular T3 supply is the normalization of hepatic alphaGDP (G-protein alpha). Following T4 supplementation at exposure to the temperature of 4 °C, Bianco showed fivefold increase in locally generated T4, while resultant T3 excess was caused by adrenergic stimulation of 5′deiodinase in BAT. A dose of T4 sufficient to provide normal T3 and alphaGDP levels was able to restore normal thermogenin activity in response to cold, while a dose of T3 sufficient to maintain UCP levels resulted in too high T3 and hepatic alpha GDP levels [38]. An adequate T3 level required for full thermogenic response can be achieved more easily through intracellular transformation of T4 to T3 rather than by T3 administration [38]. However, Beard et al. in 1984 observed the opposite results, showing that T3 infusion in iron deficiency rats could improve their ability to maintain normal body temperature, while T4 infusion failed to achieve these effects [41]. 

### 2.5. ID Can Lead to Overgrowth and Dysfunction of IBAT

In ID states, an overgrowth of IBAT is observed with paradoxical decrease in effectiveness of thermogenesis [20]. Low temperature should lead to increased levels and turnover of thyroid hormones (accelerated transformation of T4 to T3 in the liver and brown adipose tissue, higher use of T3, and increased resting metabolic rate). These processes are blunted in iron deficiency states. That is why in a rat study, T4 and T3 levels were decreased by 48% and 28%, respectively, versus controls [35]. Moreover in iron deficiency states a considerable reduction of T3 availability, and turnover was seen in brown adipose tissue [19]; and, at the temperature of 15 °C, T4 and T3 tissue utilization rates were considerably reduced (by 48% and 28%, respectively). In other studies in rats, the ability of iron deficient rats to regulate thyroid hormone levels remained impaired despite the increase in thyroid hormone levels in response to cold [35]. In a study of females with iron deficiency, 8 subjects had iron deficiency without anaemia, and 12 subjects made up the control group. The study subjects were exposed to water at a temperature of 28 °C. Women with iron deficiency appeared to have lower core body temperature and reduced oxygen consumption index at 100 min of cold exposure as well as lower T3 and T4 levels. ID women showed 20–32% higher oxygen consumption at the temperature of 36 °C, whereas their exposure to water at the temperature of 28 °C resulted in 33–81% increase in oxygen consumption. Despite the increase in metabolism, there was no parallel increase in the body’s core temperature. This finding is in line with previous observations of animals concerning IBAT overgrowth and reduced efficacy of thermogenesis [20,29]. These changes are reversible because iron supplementation significantly improved thermoregulation response as well as increased T3 and T4 levels [42]. 

### 2.6. How Iron Deficiency can Lead to Increased Senstitivty to Stress

Impaired tissue utilization of norepinephrine resulting in increased adrenergic tone, impaired response to stressors such as hypoxia and cold, and impaired synthesis and distribution of cortisol, dopamine, and serotonin are only few examples of the effects of iron deficiency. Emotional stress can lead to hyperactivity of the immune system through the secretion of cortisol [43]. Stress and emotional brain networks can lead to obesity. Stress and hypercortisolemia promote food intake and increase levels of insulin. In addition, feeding habits in which pleasurable feeding reduces stress contribute to the vicious circle of obesity [44]. Since the 17th century it has been suggested that stress plays a role in the aetiology of type 2 diabetes mellitus. Studies show that it is not only depression, but also chronic stress, anxiety, and sleep disorders that can lead to an increased risk of type 2 diabetes mellitus [45]. Too much cortisol acts on the hypothalamic-pituitary axis and impairs TSH secretion. Abnormal cortisol secretion was confirmed in iron deficiency states. Too low cortisol levels were observed as early as at one hour of cold exposure [46]. In ID histological and ultrastructural changes occur in the adrenal glands. The most affected structures are mitochondria, which often become grossly enlarged and develop unusual electron-dense inclusions. Lipid droplets in iron deficient adrenal cells were much less developed compared to cells with normal iron supply [47]. Monkeys with ID presented abnormal cortisol secretion in the social stress test, in response to an intruder and in response to pictures presenting social and unsocial issues [48]. In a study of pregnant women, higher cortisol levels were seen in ID versus control subjects [49]. Stress also increases the production of prolactin. Iron deficiency affects prolactin levels as prolactin is a peripheral marker of central dopamine release. Dopamine released from tuberoinfundibular dopaminergic neurons (TIDA) inhibits the release of prolactin from the anterior pituitary gland [50]. Studies have shown that early maternal care of offspring was associated with lower prolactin levels in later life in response to stress [51]. The most recent research shows that children born to mothers with ID had higher prolactin levels in the umbilical cord blood [52]. High prolactin levels in foetal life and childhood may set a new balance in TIDA neurons and subsequently counteract the beneficial effect of maternal care on prolactin levels, and increase the offspring sensitivity to stress in later life. 

## 3. Effect of Iron Deficiency on Autoimmune Thyroid Disease (Aitd) Development

### 3.1. AITD—The Most Prevalent Endocrinopathy

Autoimmune thyroid disease (AITD) is the most prevalent endocrinopathy and autoimmune disease all over the world. Nowadays we can say for sure that it is one of the metabolic disorders because it involves the underproduction of thyroid hormones and disturbances of the immune system. AITD is also the most frequent cause of hypothyroidism in areas of iodine sufficiency [53]. It results in the development of thyroiditis which finally in 20–30% of patients leads to hypothyroidism [54,55]. AITD affects 0.3–1.5/1000 subjects/year and is 4–10 times more frequent in women than in men (3.5–5/1000 subjects/year in women versus 0.6–1.0/1000 subjects in men [56]). AITD is diagnosed based on increased serum levels of anti-thyroid antibodies and/or hypoechogenicity of the thyroid gland at ultrasound [54,55]. Anti-thyroid peroxidase (anti-TPO) and/or anti-thyroglobulin (anti-Tg) antibodies are estimated to develop in 2–17% of women of childbearing age. To maintain normal thyroid function, a number of microelements are required. The role of iron in maintaining normal thyrometabolic status is becoming more and more recognized, and there is a growing body of evidence indicating that iron is important for the outcome of iodine prophylaxis and prevention of goitre, cognitive functions, and congenital iodine deficiency disorders [57].

### 3.2. A Link between Thyroid Disorders and Development of Obesity and Diabetes—Mutual Interactions 

Thyroid disorders and diabetes mellitus are the two most frequent endocrine comorbidities. Relationships between autoimmune thyroid disease (AITD) and type 1 diabetes mellitus (DM1) have been thoroughly studied. A growing body of evidence shows that thyroid disorders are more also frequent in patients with type 2 diabetes mellitus (DM2) [58,59,60]. Common histocompatibility antigens are considered as well as the polymorphism of genes such as CTLA4 (cytotoxic T cell antigen 4), PTPN22 (Protein Tyrosine Phosphatase Non-Receptor,) IL2Ralfa (Interleukin-2 receptor alfa), VDR (Vitamin D Receptor) and TNF (Tumour Necrosis Factor) [61]. Iodine deficiency in diabetic patients with cardiovascular risk factors contributes to the higher incidence of goitre. Regardless of their aetiology, hypothyroidism and hyperthyroidism may alter carbohydrate metabolism. Hyperinsulinism, thyroid hormone resistance, and hyperglycaemia are typical of hyperthyroidism [62]. Insulin resistance (IR), being the consequence of hyperinsulinemia and reduced utilization of glucose in peripheral tissues, is typical of hypothyroidism [63]. Metformin treatment can affect TSH suppression and alter IGF-1 (insulin-like growth factor 1) and nesfatin-1 levels, which results in thyroid dysfunction [64]. A Spanish study of 778 patients showed a positive correlation between TSH and BMI (body mass index) [65]. In a Norwegian study of 15,020 female euthyroid patients, each increase in TSH by 1 mIU/L corresponded to 0.9 kg increase in body weight and 0.3 kg/m^2^ increase in BMI. In DM increased fT3 levels have been shown due to deiodinase polymorphism. This may result from increased incidence of chronic diseases, which are seen more frequently in diabetic patients. The presence of AITD potentiates insulin resistance [66] in type 2 diabetes mellitus. There are also derangements of lipid metabolism in the hypothyroid state. Moreover hypercholesterolemia can develop as a result of decreased cholesterol clearance, reduced conversion of cholesterol to bile acids in the liver, and increased synthesis and absorption of cholesterol [67]. 

Hypertriglyceridemia, via its effect on atherosclerosis, promotes diabetic complications. Subclinical hypothyroidism can increase cardiovascular risk in diabetic patients and exacerbate diabetic nephropathy [68]. What is interesting is that we confirmed in our studies that the presence of AITD in DM1 patients reduced the risk of diabetic retinopathy, and DM1 patients with concomitant AITD had lower creatinine levels [69,70]. Thyroid hormones can affect both the distribution and volume of adipose tissue. In a study of 174 euthyroid children [71], fT4 was independently and inversely related to the volume of visceral fat. In a study of 303 people, the amount of subcutaneous fat and the ratio of subcutaneous to visceral fat was inversely related to fT4, while TSH positively correlated with the thickness of subcutaneous fat [72]. Changes in adipose tissue content may result from differences in TSH receptor and/or expression of thyroid hormone receptor in different fat depots. Expression of the TSH receptor in subcutaneous fat is higher in patients with a higher BMI value [73]. Thyroid hormone receptor expression is increased in subcutaneous adipose tissue compared to visceral adipose tissue in obese subjects versus normal weight subjects [74]. A 33% reduction in BMI after bariatric surgery resulted in 150% increase in TSH receptor expression in subcutaneous adipose tissue and a 70% increase in thyroid hormone receptor 1 expression [75].

### 3.3. How ID Can Contribute to the Increased Risk of AITD?—Possible Pathogenetic Mechanisms 

The aetiology of AITD is multifactorial and depends on complex interactions between genetic and environmental factors leading to the breakdown of immune tolerance [76]. Genetic factors are estimated to contribute to the disease in 70% of cases, while environmental factors associated with the activation of the innate immune system in 20–30% [77]. In most cases of autoimmune polyendocrine syndrome (APS), AITD appears to be the first manifestation of the disease [78]. Studies confirm that AITD patients are more likely to suffer from Huntington’s disease, multiple sclerosis, coeliac disease, diabetes, sickle cell anaemia, sarcoidosis, alopecia areata, rheumatoid arthritis, polymyalgia, articular psoriasis, scleroderma, and the hepatitis C virus (HCV) cryoglobulinemia [79]. 

The relationship between iron deficiency anaemia (IDA) and autoimmune disorders is more pronounced in women. Its risk factors include arterial hypertension, dyslipidaemia, cancer, allergic rhinitis, chronic obstructive pulmonary disease (COPD), urticaria, and chronic liver disease. Another important risk factor is the patient’s age, with the highest risk observed in subjects aged 20–40. In the group of patients aged 20–40 years, approximately 65% are likely to develop an autoimmune disease within 5 years of being diagnosed with IDA [80]. In a study of 2581 pregnant women, anti-TPO antibody level was higher in subjects with iron deficiency, with no differences in total thyroxine (TT4) levels between study groups. Hypothyroidism and subclinical hypothyroidism were more prevalent in IDA group versus mild or no iron deficiency groups. The study has shown that the incidence of AITD was growing along with the severity of iron deficiency. In a meta-analysis by Luo et al. from 2021, ID in women of a reproductive age resulted in a twofold increase in the risk of elevated titres of TPO- and/or Tg-antibodies [81]. 

One of the hypotheses that explain the increased incidence of autoimmune disorders in the 21st century says that excessive hygiene lowers the threshold for immune activation [82] in response to pathogens. Numerous autoimmune diseases are triggered by infection. Reducing inflammation in autoimmune diseases can restore the immune balance and induce remission. Infections can precede rheumatic fever [83], Guillain-Barre syndrome [84], and borreliosis [85]. A connection has been proven between Epstein–Barr virus (EBV) infection and systemic lupus erythematosus, while endemic malaria has been shown to induce antinuclear antibodies (ANA). H. pylori infection is associated with the development of autoimmune gastritis [86]. Infections may result in disruption of the immune regulation, which may cause autoantigen mimicry. Infection can also affect target organs and increase their susceptibility to autoimmune disorders. The dual role of infection means that it is both an indicator of autoimmunity and a promoter of disease progression and escalation of the autoimmune process to the autoimmune disease. Molecular mimicry, antigen expression, and tissue modifications can lead to antigen-specific signals that initiate autoimmune reactions. Other effects include the induction of the pathogen associated molecular pattern (PAMP) and toll-like receptors involved in the progression from the autoimmune reaction to the autoimmune disease. Infection can shift the body’s normal immune response toward a pathological process that will result in a full-blown disease. Effective apoptosis is a mechanism that enables the body to eliminate damaged cells without inducing inflammatory state. Infection and other environmental factors can impair apoptosis and increase the risk of autoimmune process. Damaged cells, that are not cleared by macrophages and monocytes, will remain in tissues as a potential source of the autoimmune process. Impaired apoptosis can lead to autoimmune lymphoproliferative disorders associated with cluster of differentiation 95 (CD95) (fas)-fas-ligand mutation. Another example is neonatal lupus erythematosus (a disorder accompanied by congenital heart block), caused by placental transfer of anti-Ro/SSA and anti-La/SSB antibodies. Iron deficiency can impair ferroptosis—a type of iron-dependent cell death. Iron is necessary for immune response reactions. Mutation in the gene of transferrin receptor 1 (TRF1), a protein responsible for iron transport into the lymphocytes, can lead to immune deficiency states in humans, with low IgG levels and reduced B cell and T cell proliferation [87]. Iron is of key importance for appropriate activities of peroxidases and synthases involved in nitric oxide generation, which is crucial for maintaining proper functions of immune cells. Moreover iron is involved in regulation of cytokine production and signalling pathways [88]. ID can affect cytokine expression on lymphocytes, leading to an increase in the number of cells with interferon gamma (IFN-gamma) expression and a decrease in cells with interleukin 4 (IL-4) expression [89]. Iron is also involved in activation of NF-kappaB (nuclear factor kappa-light-chain-enhancer of activated B cells), transcription factors necessary for the expression of genes crucial for innate and adaptive immunity [90]. Iron deficiency can alter signalling via the toll-like receptor 4 (TLR4) pathway [91] and affect the activity of iron-dependent ubiquitin ligase [92] and regulation of SHP1 (Src homology region 2 domain-containing tyrosine phosphatase) [93]. Adequate cell-mediated immune response is delayed in ID [94]. A number of studies in humans have shown an impairment of innate immunity in subjects with ID. Bactericidal activity of macrophages is reduced, while neutrophils contain lower amounts of myeloperoxidase (MPO) involved in the generation of reactive oxygen species and responsible for intracellular pathogen death [95]. Children with ID had lower levels of IgG and interleukin 6 (IL-6) and decreased phagocytic activity with reduced oxidative burst in neutrophils [96]. ID affects blastogenesis and mitogenesis of T cells and protein kinase C activity [97]. In mice it can lead to a decrease in a T cell-mediated antigen specific response, production of antibodies, and B cell proliferation [98]. Moreover, there is a reduction in induction of cyclin S, and delayed entry of B cells in phase S of the cell cycle during B cell proliferation is observed [99]. Histone demethylation is of key importance for T cell differentiation. Demethylation of cyclin E1 of histone H3K9 is iron dependent, which in case of iron deficiency may impair T lymphocyte differentiation [100]. The most severely affected in ID was TLR-dependent B lymphocyte proliferation, followed by B lymphocyte response to BCR (b-cell receptor) stimulation and the relatively least affected T lymphocyte response to TCR (T-cell receptor) stimulation [99]. 

## 4. The Impact of Iron Deficiency on Obesity and Diabetes

### 4.1. How Iron Deficiency can Contribute to the Development of Obesity—General Interactions

Globally about 2 billion people are overweight and 650 million people are obese [101]. These numbers have increased threefold since 1975. The most terrifying data has been reported for children and adolescents—almost 380 million of them are overweight or obese. Obesity is not just a statistic. It results in numerous diseases including first of all diabetes and its complications. At the end of the 20th century diabetes affected 151 million people worldwide [102] and for the next 2 decades this number has increased up to 463 million [103]. Of the new cases of diabetes, 80% are seen in countries of low and medium income [103]. 

Obesity and iron are inextricably linked. Obesity correlates with the severity of inflammation, which is controlled mostly by CD4+ T lymphocytes; therefore, there is an increase in IL-6 secretion, which makes CD4+ cells differentiate to Th17. All of this results in increased secretion of proinflammatory adipokines: leptin, resistin, lipocalin 2, visfatin, monocyte chemoattractant protein, and TNF alpha [104]. Population studies have shown that ID is considerably more prevalent in obese subjects versus subjects with normal body weight [105]. Iron deficiency more frequently affects obese people. In the *NHANE-I* study, Micozzi et al. [106] found that higher BMI values were associated with lower iron and transferrin levels. Obese versus nonobese adults were reported to have lower iron, transferrin saturation, and MCV, as well as a considerably lower soluble transferrin receptor (sTfR). Ferritin and C-reactive protein (CRP) levels were elevated in obese subjects and showed a positive correlation with BMI [107]. The hypotheses proposed to explain these findings include higher blood volume in obese subjects [108], chronic inflammatory state resulting in alterations in iron sequestration, and effects of hepcidin and lipocalin 2—a protein with increased expression in white fatty tissue, which acts as siderophore binding protein (SBP) and is upregulated in inflammation, with resultant impairment of iron absorption [109]. Obesity can also change iron levels in the brain—mesencephalon and thalamus. At the same time, an increase in neurodegeneration markers (alpha-synuclein, F2-isoprastans) is observed in the midbrain [110]. A diet rich in iron resulted in body weight reduction in obese mice with DM2 [111]. ID can promote obesity also by increasing fatigue, which results in lower physical activity [112]. Impairment of the mitochondrial respiratory chain can reduce exercise capacity and potentiate insulin resistance [113]. ID can lead to oxidative stress with subsequent increase in proinflammatory cytokines, which by activating leucocytes can promote fat accumulation [114]. In other studies ID caused a more pronounced body weight gain and insulin resistance by reducing AMP (adenosine monophosphate)-activated protein kinase activity [115].

### 4.2. Iron Effect on Adipose and Muscle Tissue Metabolism

#### 4.2.1. Iron Deficiency versus Adipose Tissue Metabolism and Interactions with Muscle Tissue 

Adipose and muscle tissues come to mutual interactions, and iron can be involved in these interactions.

In the human body there are three major types of adipose tissue: non-thermogenic white adipose tissue and two types of thermogenic adipose tissue, constitutively brown and inducible beige adipocytes [116]. The white adipocytes store excess energy in triglycerides [116], while the brown and beige adipocytes convert energy to heat [117]. The white adipose tissue (WAT) contains large and unilocular lipid droplets, while the brown adipose tissue (BAT) and beige adipose tissue (BeAT) contain small and multilocular lipid droplets required for beta-oxidation [117]. Interscapular brown tissue (IBAT) is found predominantly in the interscapular, subscapular, axillary, cervical [118], aortic, paravertebral, and suprarenal locations (Figure 1). Beige precursor cells originate from the same white adipocyte lineage—endothelium and perivascular cells within WAT depots [119], while brown precursor cells share the same lineage with myocytes [120]—and they originate from myogenic factor 5 (MYF5+) mesenchymal stem cells in the mesoderm. The amount of BAT is inversely related to the risk of obesity and IR [121], while more WAT is associated with increased cardiovascular risk [122]. WAT and BAT communicate with other organs via adipokines, batokines, lipokines, and exosomal miRNA [123,124] (Figure 2). Adiponectin and leptin 2 are the two major adipokines that affect growth and function of the muscles [125]. Other adipokines secreted by WAT include vascular endothelial growth factor A (VEGFA), osteopontin, resistin, omentin, cardiotropin 1, apelin, and plasminogen activator inhibitor 1 (PAI-1) [126]. By secreting VEGFA, the adipose tissue affects angiogenesis, which is crucial for normal muscle growth and differentiation. In addition to interleukin 8 (IL-8) and angiopoietin 1, VEGFA is the best described factor involved in the regulation of exercise-induced angiogenesis in the muscles [127]. Adiponectin plays important anti-inflammatory, antidiabetic, antihypertrophic, and antiatherogenic functions [125]. It affects muscle development, regeneration and work regulation [128]. Leptin stimulates monocyte growth via the activation of peroxisome proliferator-activated receptor gamma coactivator (PCG) 1-alpha secretion and miogenin, as well as via the inhibition of myocyte growth suppressors (myostatin and dystrophin) [129]. Furthermore, via the secretion of retinol binding protein 4 (RBP 4), wisfatin, vaspin, chemerin and, omentin, the adipose tissue regulates muscle glucose uptake [126]. BAT and BeAT secreted bone morphogenetic protein 8b (BMP8b), fibroblast growth factor 21 (FGF21), endothelin 1, IL-6, ependymin related 1 protein (EPDR1), and lipocalin type prostaglandin D synthase (LPGDS) [124]. FGF21 can stimulate thermogenesis, by increasing the formation of mitochondrial cristae in the muscles in response to cold exposure [130]. Palmitoleate improves glucose homeostasis and insulin sensitivity by increasing glucose uptake, reducing inflammation in the muscles and reducing TAG accumulation [131]. Lipokine 12,13-diHOME, produced in BAT at exposure to cold and exercise [132], stimulates lipid uptake in the muscles and potentiates mitochondrial oxidation. Several adipose-derived exosomal miRNAs can be transferred to muscle cells, where they improve glucose tolerance via repression of PPARy (miR-27a and miR155) [133]. In addition to the three main types of adipose tissue, there is also intramuscular adipose tissue (IMAT), epicardial WAT, and pink adipose tissue (PINK). IMAT includes adipocytes located between muscles and perimysial adipocytes located within a single muscle fibre. IMAT makes from 28% to 82% of the visceral adipose tissue. The amount of IMAT is positively correlated with IR and cardiovascular risk. Due to its proximity to the muscles, IMAT is highly sensitive to myokines, which cause browning of the adipose tissue, similar to that caused by musclin. Epicardial WAT, due to its location nearby the heart and coronary vessels, constitutes a source of proinflammatory factors: leptin, Il-1beta, IL-6, and CSF-6 [134]. PINK adipocytes occur in the breast during pregnancy and lactation, being transformed from the subcutaneous WAT [135]. What is interesting is that during the involution of the mammary gland, some of the epithelial cells in the front of the subcutaneous tissue are browning [136]. Mitochondria [137] in the IBAT have different morphology when compared to IWAT—they are bigger and contain more mitochondrial cristae. It is the content of the haem cofactors of mitochondrial enzyme cytochrome oxidase (CIV) that gives the tissue its brown colour and its name [138]. Progenitors of thermogenic cells differ from white adipocyte precursors in their ability to store iron, which is more pronounced in brown adipocyte stem cells [139]. Iron demands in brown adipocytes are higher versus white adipocytes because of its utilization as porphyrin iron and iron-sulphur clusters in the mitochondrial enzymes. Festa et al. [140] have shown that iron metabolism-related genes are up-regulated during adipocyte differentiation. TfR1 expression was increased in IBAT, and iron content in IBAT was threefold higher than in WAT [118]. Browning of IWAT is associated with increased iron regulatory protein 2 (IRP2) and TfR1 and UCP1 [118]. Increased iron influx into the mitochondria is essential for thermogenic adipocyte differentiation. Mitochondrial biogenesis is a key metabolic process for adipocyte differentiation. Knockdown of mitoferrin 1 and 2, the two proteins involved in iron storage in the mitochondria, results in lower mitochondrial iron content in adipocytes, which, by reducing mitochondrial oxygen use and intracellular ATP levels, leads to lower expression of adipogenic genes and reduced synthesis of lipids during adipocyte differentiation [141]. Other studies confirm that suppression of cellular iron import results in impairment of mitochondrial biogenesis and thermogenic function of the adipose tissue [141]. In a mice study, iron storage was higher at exposure to cold (22° degrees) versus neutral temperature (30°). In ID an overgrowth of IBAT is observed with paradoxical decrease in effectiveness of thermogenesis [20]. Low temperatures should increase thyroid hormone levels and turnover (acceleration of T4 to T3 conversion in the liver and IBAT and higher T3 utilization). These processes are blunted in ID [35]. 

#### 4.2.2. Effects of Iron Deficiency on Muscle Metabolism

Blood levels of oxygen, transported with haemoglobin, and cardiac output are the key elements crucial for muscle work. Baynes and Bothwell 1990 [142] have shown that lethargy, apathy, and little exercise are frequently seen in ID. Dallman et al. have found a decrease in oxygen delivery and muscle oxidative capacity [143]. Moreover, there is a lower content of the mitochondrial iron-sulphur clusters [144], cytochrome enzymes [145], and total mitochondrial oxidative capacity in the muscles [146]. In ID, the ability to exercise is impaired by approximately 80% versus control group [147]. Iron treatment resulted in improvement of these disorders as early as within 4 days. Also aconitase, a protein containing iron-sulphur clusters, is highly sensitive to ID [147]. This enzyme, which is of key importance for gluconeogenesis, restores its activity as soon as within 15 h of iron deficiency correction [148]. Gluconeogenesis markers in ID are increased in order to ensure appropriate glucose supply to tissues [149,150]. In the study by Celsing et al., lactate levels were increased in study patients with ID, while oxygen consumption in tissues was decreased [151]. In Edgerton’s study, oxygen transport was reduced by 15% during maximal exercise in subjects with ID, whereas iron treatment was able to restore normal physical performance [152]. Perkkio et al. have found that malate and pyruvate oxidase activity in the muscles was reduced by 35%, and increased to nearly 85% of normal within 10 days of iron supplementation [153]. Iron supplementation also improves resting metabolic rate (RMR). A study of two female athletes with ID supplemented with iron has shown an improvement of their thyrometabolic status and RMR at 8 and 16 weeks of the treatment [154]. 

There is no doubt that muscles are the largest gland in the human body. Myocytes actively produce more than 300 myokines [155,156] (Figure 3), which in response to muscle contraction, via para- or endocrine mechanisms, can affect metabolism in virtually all organs in the body, including the fatty tissue, liver, pancreas, bones, and brain. We propose to name the study of muscles as an endocrine organ with the term of ”myocrynology”. Muscles can protect against inflammation, IR, hyperlipidaemia, DM2, and cancer. Furthermore, myokines can modulate the hypothalamic-pituitary axis, which shows the influence of muscles on the neuroendocrine system [157]. Physical exercise can suppress hunger, which is associated with increased levels of IL-6, irisin, nesfatin 1, and ghrelin [158]. Elevated Il-6 levels in the lateral parabrachial nucleus (LPBN) decrease food intake and intensify thermogenesis in IBAT, which suggests that myokines act as obesity regulators in the brain [159]. Physical exercise results in increased synthesis of ketone bodies (mainly acetoacetate and D-beta-hydroxybutyrate (DBHB)). DBHB can cross the blood–brain barrier, accumulates in the hippocampus, and stimulates histone acetylation in the brain-derived neurotrophic factor (BDNF) promotor region, which results in increased BDNF expression [160]. Noteworthy is that in the iron deficiency states, ketogenesis is impaired, with subsequent reduction in citrate synthase and succinate dehydrogenase activities and impaired production of free fatty acids (FFA) and ketone bodies [161,162]. Signals mediated by myokines are essential to proper body metabolism. Secretion of myokines, myomiRs, and factors contained within exosomes occurs in response to low glycogen in muscle cells [163] at contraction. Physical exercise modulates the muscle–adipose crosstalk and results in a reduction of inflammation, a reduction of the risk of sarcopenia, and a decrease in body fat accumulation. IL-6, IL-15, irisin, myostatin, myonectin, fibroblast growth factor (FGF21), and musclin, secreted in the muscles, exert their effects in the adipose tissue and regulate thermogenesis, lipolysis, glucose metabolism, and fatty acid uptake [164]. Irisin affects adipogenesis [165], stimulates WAT browning, and increases the expression of thermogenesis-related genes in beige adipocytes [165]. Musclin, released during exercise, activates peroxisome proliferator-activated receptor gamma (PPAR-gamma)-mediated WAT browning [140]. Myonectin administered in mice resulted in a decrease in fatty acid levels [166] and increased free fatty acid uptake by adipocytes and hepatocytes [166]. In mice studies, overexpression of IL-15 reduced the risk of obesity [167]. Exercise improves beta cell function in patients with prediabetes and diabetes [168]. The benefits of physical activity in diabetes mellitus are not only the consequence of increased energy expenditure, but also come from pluripotential effects, which have not been fully elucidated [169]. IL-6 stimulates insulin secretion by beta cells [170] and improves myocyte glucose uptake [171]. Intraperitoneal administration of irisin in rats with DM2 reduces body weight, lowers fasting blood glucose, and improves glucose tolerance [172]. Natalichccio et al. have shown that increased levels of irisin prevent palmitate-induced apoptosis of beta cells [27]. The beta cell effects can be mediated by muscle metabolites: beta-aminoisobutyric acid (BAIBA) and 3-hydroxyisobutyrate (3-HIB) [173]. BAIBA stimulates adipocyte leptin secretion [148] and increases beta oxidation in the muscles. Pancreatic beta cell function can also be regulated by the expression of miRNA (miR-1, MiR-133a, miR-23b, miR-29, and miR-206) in myocytes called myomiRs [174,175]. Dysfunction of the muscle tissue in iron deficiency states can result in a decrease in myokine secretion, which counteracts the beneficial effects of exercise on the human body, thereby increasing the risk of obesity and DM2.

### 4.3. Effects of Iron Deficiency on the Development of Type 2 Diabetes Mellitus

DM2 is often caused by excess fat tissue. The higher the degree of obesity, the higher the percentage of people with diabetes. Metabolically obese (visceral obesity with normal weight) represents fat levels that are higher than average—even if a person’s weight is at or below the standard for their height, they are also at greater risk of diabetes. Iron is an important regulator of glucose metabolism. The relationship between iron metabolism status and diabetes is complex and has not been fully elucidated. A number of papers have shown a relationship between ferritin levels and increased risk of diabetes mellitus (DM) [176,177] and insulin resistance (IR) [178]. Iron can block suppression of glucose production by insulin in the liver, while insulin stimulates ferritin synthesis [179] and redistributes transferrin receptors on the cell surface [180]. Fleming has found that numerous genes involved in iron metabolism are changed in DM [181]. Iron deposition in muscles reduces the uptake of glucose [182] and stimulates the expression of DMT1 in the pancreas [183]. Instead, iron deficiency can promote the expression of insulin receptor and glucose transporter type 4 (GLUT4) transcription in muscles [184]. Some studies focus on the inflammatory state induced by iron overload and reactive oxygen species (ROS) formation, which can affect insulin resistance [185]. Another possible mechanism is the development of IR as a result of brain iron overload [186]. By impairing the mitochondrial respiratory chain, ID can reduce exercise capacity and potentiate insulin resistance [113]. Although in iron deficiency anaemia (IDA) glucose and lactate levels are elevated, glucose compensatory utilization is also increased as a result of improved insulin sensitivity [187]. These changes involve modification of the insulin receptor and changes in glucose transportation. A study by Kemp et al. has shown an increase in gluconeogenesis in ID states, which is supposed to provide an adequate glucose supply to the tissues [149]. In a study by Wasserman et al., levels of glucagon, cortisol, epinephrine, and norepinephrine were increased in ID dogs during their exercise [188]. Ohira et al. observed higher blood glucose levels in animals with ID, as well as increased muscle activity of lactate dehydrogenase in these animals [187]. The most recent studies investigate the effect of iron metabolism on diabetes via human microbiome modifications [98,189]. In ID, cell sensitivity to bacterial endotoxins is increased [190]. As a result of intestinal bacterial fermentation, short chain fatty acids (SCFAs) are produced. SCFAs bind to and activate G-protein coupled receptors (Gpr-1) on enteroendocrine cells and regulate secretion of glucagon-like peptide 1 (GLP1) and peptide-YY (PYY) [191]. By binding to Gpr-1, endotoxins stimulate pancreatic beta-cell receptors and inhibit insulin secretion [192]. Moreover, microbiome stimulates bile acid secretion, whereas bile acids can impair insulin production [193]. ID impairs metabolic control in diabetes and increases the frequency of diabetic complications [194,195,196,197,198]. Therefore, preventing and treating ID and IDA provides significant benefits to DM patients.

To summarize, IBAT has the ability to provide thermogenesis and protects against obesity by clearing triglycerides, releasing batokines, and mitigating IR. WAT stores excess energy and secretes endocrine factors that lead to IR. Browning of WAT can be caused by exercise, exposure to cold, metformin, beta-aminoisobutyric acid (BAIBA), gamma-aminobutyric acid (GABA), PPAR-gamma agonists, Janus kinase (JAK) inhibition, and irisin [136,199] (Figure 4). ID can impair this beneficial process. By affecting the muscle and adipose tissue functions, iron deficiency can alter endocrine activity of these tissues and interfere with their mutual interactions. Mitochondrial impairment in IBAT in iron deficiency, along with ineffective thermogenesis in the mechanism of abnormal thyroid hormone secretion, can lead to the development of obesity and type 2 diabetes mellitus.

## 5. Conclusions

Iron deficiency anaemia remains a major health problem all over the world, and is among the main causes of mortality and morbidity. However the aim of our paper was to emphasize some other risks associated with iron deficiency, of which physicians and patients are less aware, and which include metabolic disorders: obesity and diabetes. The lifestyle in the 21st century is inevitably associated with a sedentary way of living and physical and emotional stress, which puts us at an increased risk of metabolic disorders including obesity, diabetes, and thyroid dysfunction. Iron deficiency leads to disorders in the functioning of the immune system and impairs the body’s response to emotional and physical stress, increasing the risk of diabetes and obesity. An important role in the development of metabolic disorders in iron deficiency states can be played by thermoregulation disturbances. Iron plays an extremely important role in the cross-talk between lipocrinology, glucocrinology, and myocrinology, with thyroid hormones acting as a non-passive bystander. The discovery made 15 years ago that skeletal muscle is an endocrine organ opened a new paradigm that we propose to name “myocrynology”. In conclusion, iron is essential for proper functioning of all cells in the human body including endocrine organs, and is involved in the complex interplay of hormonal signals, also on the cellular level (Figure 5). ”Ferrocrinology” is the term proposed by us to describe the study of the effects of iron on metabolism disorders, ednocrinopathies, and mutual interactions of adipose and muscle tissues, which are the largest endocrine organs in the human body (Figure 6).

## Figures and Tables

**Figure 1 nutrients-14-04693-f001:**
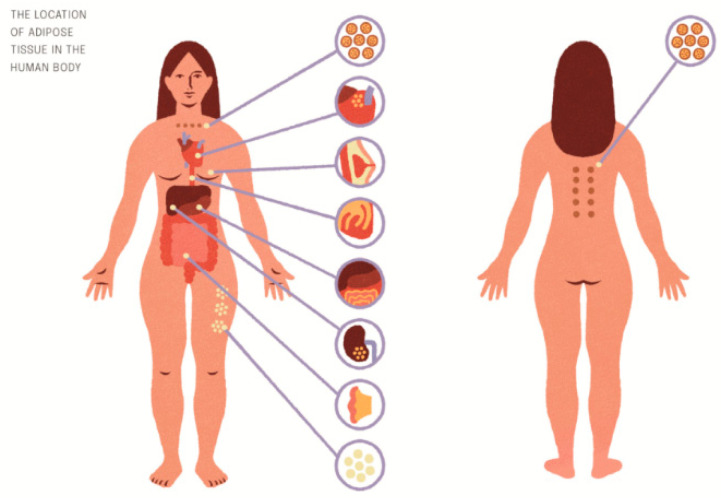
The location of adipose tissue in the human body. In the human body there are three major types of adipose tissue: nonthermogenic white adipose tissue and two types of thermogenic adipose tissue: constitutively brown and inducible beige adipocytes. In the figure we have also presented another types of adiose tissue: epicardial WAT and pink adipose tissue (PINK).

**Figure 2 nutrients-14-04693-f002:**
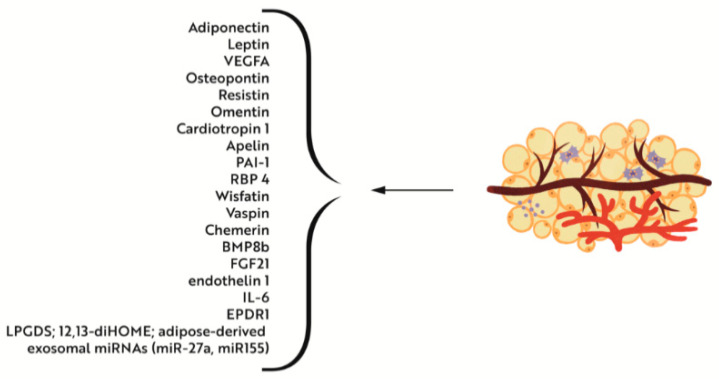
Lipocrinology—adipose tissue as an endocrine organ is able to secrete a lot of hormones with crucial metabolic effect.

**Figure 3 nutrients-14-04693-f003:**
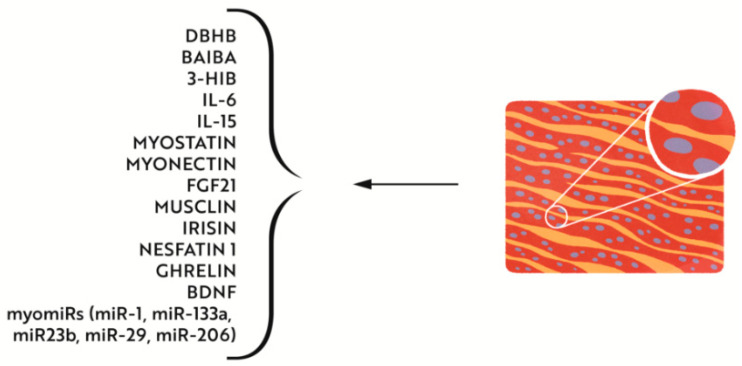
“Myocrinology”—muscle tissue as an endocrine organ.

**Figure 4 nutrients-14-04693-f004:**
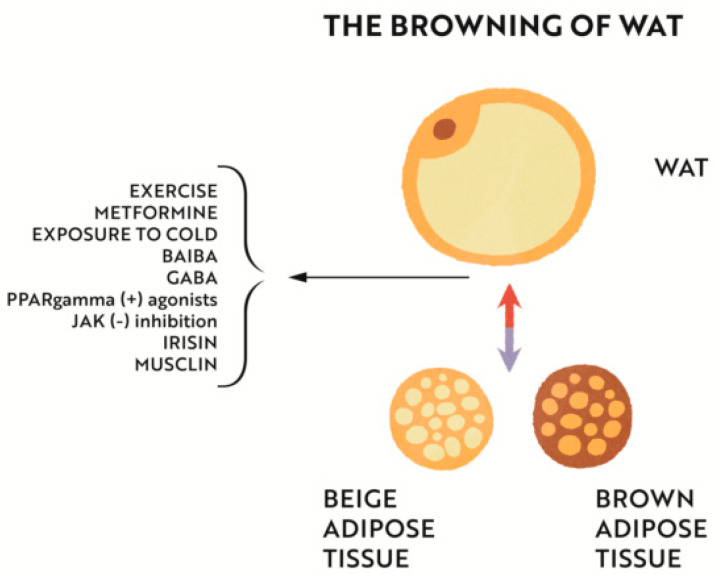
Browning of WAT can be caused by: exercise, exposure to cold, metformin, beta-aminoisobutyric acid (BAIBA), gamma-aminobutyric acid (GABA), PPAR-gamma agonists, Janus Kinase (JAK) inhibition, and irisin.

**Figure 5 nutrients-14-04693-f005:**
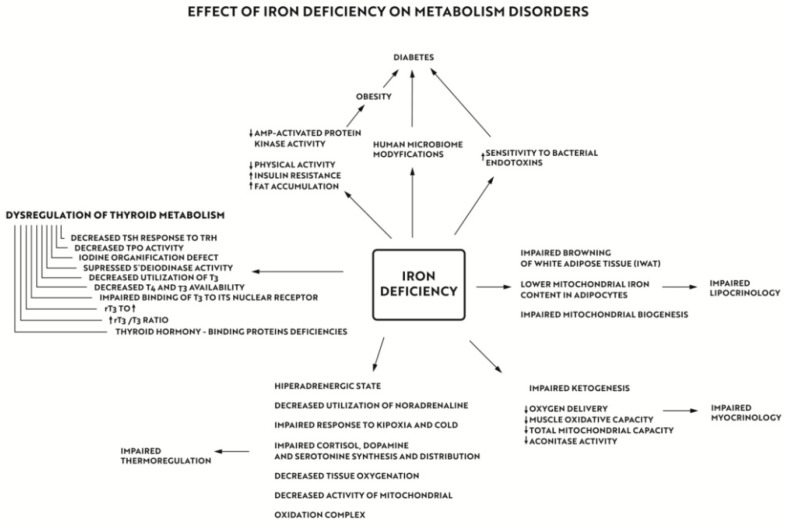
The effect of iron deficiency on metabolism disorders. Iron plays an extremely important role in the cross-talk between lipocrinology, glucocrinology and myocrinology, with thyroid hormones acting as a non-passive bystander and it’s effect on thermoregulation.

**Figure 6 nutrients-14-04693-f006:**
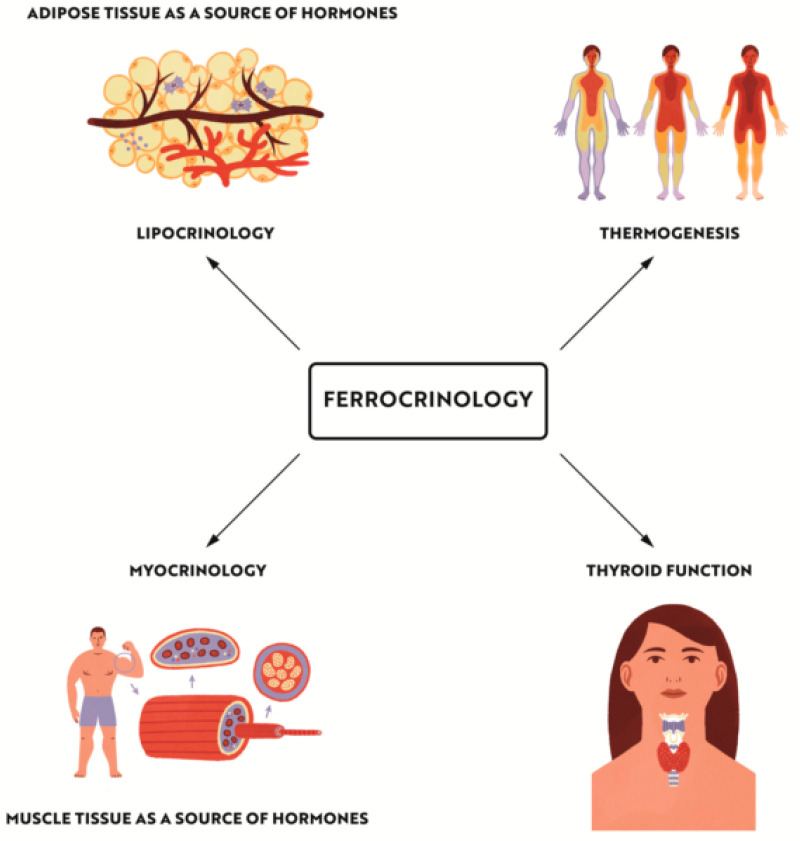
“Ferrocrinology” is the term proposed by us to describe the study of the effects of iron on metabolism disorders, endocrinopathies as AITD and mutual interactions of adipose and muscle tissues with it’s influence on thermoregulation disturbances.

## Data Availability

Study did not report any data.

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
