# Peer review of "“Ferrocrinology”—Iron Is an Important Factor Involved in Gluco- and Lipocrinology"

_nutrients, 2022, doi:10.3390/nu14214693_

Round 1
Reviewer 1 Report
First, the ubiquitous improper use of commas (or lack thereof), semicolons, etc. makes reading this manuscript more difficult than necessary. Additionally, the lack of identification of abbreviations at first use (if at for many of them) makes this manuscript impossible to be followed by a non-expert in the field. Also, formatting must be checked throughout, including the numerous errors in the references. Far too many sentences start "there are", "it is", etc.
The statement leading off the third paragraph of section 2 that "there is no type 2 diabetes (DM 2) without obesity" is simply not true.
The paragraphs tend to be huge and rambling without clear flow leading directly from the medical condition to the role of iron.
Figures of suggested pathways showing points of direct involvement of iron or an iron-dependent process would be most helpful.
The authors need to take their massive research and start writing from scratch from a clear outline with aid from someone with better English grammar skills.
Author Response
We have sent responses in the attachment.
Dear Reviewer,
We sincerely thank you for yours useful suggestions.
RESPONSE 1
„First, the ubiquitous improper use of commas (or lack thereof), semicolons, etc. makes reading this manuscript more difficult than necessary. Additionally, the lack of identification of abbreviations at first use (if at for many of them) makes this manuscript impossible to be followed by a non-expert in the field. Also, formatting must be checked throughout, including the numerous errors in the references. Far too many sentences start "there are", "it is", etc. „
It was corrected according to your suggestions.
RESPONSE 2
"The statement leading off the third paragraph of section 2 that "there is no type 2 diabetes (DM 2) without obesity" is simply not true"
We have removed that statement.
RESPONSE 3
"The paragraphs tend to be huge and rambling without clear flow leading directly from the medical condition to the role of iron."
It was corrected according to your suggestions.
RESPONSE 4
“Figures of suggested pathways showing points of direct involvement of iron or an iron-dependent process would be most helpful.”
We have added few orignal figures.
RESPONSE 5
"The authors need to take their massive research and start writing from scratch from a clear outline with aid from someone with better English grammar skills."
We have improved english in our work with cooperation with english native speaker.

Reviewer 2 Report
Comments:
Q1: Abstract section was suggested to be further improved, to make a better summary of the full text.
Q2: The too many paragraphs were suggested to be further subdivided with subtitles or reorganized to improve the presentations.
Q3: Many paragraphs in the manuscript did not involve statements related to iron metabolism. What is the relationship between these passages and the central theme of this paper?.
Q4: In this paper, I am not sure that the new term proposed could accurately summarize the meanings. Like Ferrocrinology, is iron secreted by cells in the body? Is this article focusing on the various functions of iron in the body?.
Q5: The schematic diagrams of the core mechanisms of “the role of iron in the cross-talk between glucocrinology, lipocrinology and myocrinology, with thyroid hormones acting as an active bystander” are suggested to be supplemented, which would be better and more intuitive for readers understanding.
Q6: What are the biggest health risks associated with iron deficiency? Anemia? Or obesity, abnormal glucose and lipid metabolism mentioned in this article?
Q7: The effective summary of the reported literatures needs to be further improved.
Author Response
We have sent responses in the attachment.
Dear Reviewer,
We sincerely thank you for yours useful suggestions.
RESPONSE 1
„Abstract section was suggested to be further improved, to make a better summary of the full text.”
It was corrected according to your suggestions.
Ferrocrinology is the term proposed by us to describe the study of iron effects on the functioning of adipose tissue, which together with muscle tissue makes the largest endocrine organ in the human body. By impairing exercise capacity, reducing AMP-activated kinase activity, and enhancing insulin resistance, iron deficiency can lead to the development of obesity and type 2 diabetes mellitus. Due to impaired browning of white adipose tissue and reduced mitochondrial iron content in adipocytes, iron deficiency can cause dysfunction of brown adipose tissue. By reducing ketogenesis, aconitase activity, and total mitochondrial capacity, ID impairs muscle performance. Another important aspect is the effect of iron deficiency on the impairment of thermogenesis due to reduced binding of thyroid hormones to their nuclear receptors, with subsequently impaired utilization of norepinephrine in tissues, and impaired synthesis and distribution of cortisol, which all make the body’s reactivity to stress in ID more pronounced. We also describe how iron deficiency can lead to the development of the most common endocrinopathy, autoimmune thyroid disease. In this paper, we have discussed the role of iron in the cross-talk between glucocrinology, lipocrinology and myocrinology, with thyroid hormones acting as an active bystander.
RESPONSE 2
“The too many paragraphs were suggested to be further subdivided with subtitles or reorganized to improve the presentations.”
It was corrected according to your suggestions.
RESPONSE 3
„Many paragraphs in the manuscript did not involve statements related to iron metabolism. What is the relationship between these passages and the central theme of this paper?”
It was corrected according to your suggestions – we have added these statements.
RESPONSE 4
We understand your suggestion about the title, nevethless we have written explanation why „Ferrocrinology” is best name in our opinion.
In conclusion, iron is essential for proper functioning of all cells in the human body including endocrine organs, and is involved in the complex interplay of hormonal signals, also on the cellular level.
RESPONSE 5
„The schematic diagrams of the core mechanisms of “the role of iron in the cross-talk between glucocrinology, lipocrinology and myocrinology, with thyroid hormones acting as an active bystander” are suggested to be supplemented, which would be better and more intuitive for readers understanding. „
We have added few original figures.
RESPONSE 6
„What are the biggest health risks associated with iron deficiency? Anemia? Or obesity, abnormal glucose and lipid metabolism mentioned in this article?”
Iron deficiency anaemia remains a major health problem all over the world, and is among the main causes of mortality and morbidity. However the aim of our paper was to emphasize some other risks associated with iron deficiency, of which physicians and patients are less aware, and which include metabolic disorders: obesity and diabetes.
RESPONSE 7
“The effective summary of the reported literatures needs to be further improved”
We have modified “Conclussions” section.
Kind regards,
Szklarz Michał

Round 2
Reviewer 1 Report
The manuscript is significantly improved; however, the same problems exist, just not to the same degree. For example, the abbreviation ID is used in the abstract without being defined.
Continued improvement in English grammar is required. One only has to get to the third sentence of the introduction for the first major error. Why is the title in quotation marks? Why is the first quotation mark subscripted?
Use of first-person voice should be removed from abstract.
Fonts change from reference to reference.
Major editing is needed.
Text associated with figures is too small to be read.
Author Response
Dear Reviewer,
We sincerely thank you for yours useful comments – we have changed title, abstract and figures according to your suggestions.
Kind regards,
Szklarz Michał

Reviewer 2 Report
Thanks for these modifications, I have no other comments.
Author Response
Dear Reviewer,
We sincerely thank you for your suggestions in whole review of our manuscript.
Kind regards,
Szklarz Michał